# Amazing epsilon-shaped trend for fretting fatigue characteristics in AM60 magnesium alloy under stress-controlled cyclic conditions at bending loads with zero mean stress

**Saeid Rezanezhad, Mohammad Azadi** ⬤ *

Faculty of Mechanical Engineering, Semnan University, Semnan, Iran

* m_azadi@semnan.ac.ir

**Data Availability Statement:** All relevant data are within the paper.

## Abstract

In the present article, fatigue properties (pure and fretting) of magnesium alloys (AM60) under cyclic bending loading were compared. For this objective, a rotary fatigue testing device was utilized with a fretting module on standard cylindrical samples under bending loads with zero means stress. The fretting fatigue condition decreased fatigue lifetime compared with pure fatigue but in an amazing Epsilon-shaped trend. Comparatively speaking to the state of pure fatigue, the fatigue lifetime of the fretting fatigue condition reduced by 91.0% and 44.8%, respectively, between the lowest level of stress (80 MPa) and the greatest level of stress (120 MPa). To study the fracture behavior and the fractography analysis, field-emission scanning electron microscopy (FESEM) was utilized. In general, since both quasi-cleavage and cleavage were seen; therefore, the fracture behavior for all samples was brittle. In both test conditions (fretting fatigue and pure fatigue), at higher stress levels, the average crack length was higher than at low-stress levels. In addition, the number of cracks (in high- and low-stress levels) was observed to be less in fretting fatigue conditions than in pure fatigue conditions, but the average crack length in fretting fatigue conditions in high-stress levels and low-stress levels was 212.82% and 259.47% higher than the average crack length under the pure fatigue condition, respectively.

## 1. Introduction

When interacting entities or surfaces are subjected to vibratory loads or fluctuating stresses, fretting frequently occurs as a small relative tangential motion [1, 2]. In this instance, the fretting force (or the contact stress) causes cracks to start and grow, which is how the fretting fatigue phenomenon emerges [3]. The phenomena of fretting fatigue in vehicle engine parts, which occurs most often when the first piston ring and the top ring groove come into contact, might be explained by taking into account vibration friction wear and cyclical mechanical loads [4]. Researchers should take fretting fatigue behavior into account to increase component reliability by taking into account industrial requirements [5]. In magnesium alloys, high

**Funding:** The authors received no specific funding for this work.

**Competing interests:** The authors have declared that no competing interests exist.

specific strength, outstanding castability, exceptional machinability, and high stiffness make them excellent candidates for enhancing vehicle fuel economy. Due to this, magnesium alloys are utilized in a variety of industries, including automobile [6, 7], aerospace, and others [8, 9]; AM60 is one of the magnesium alloys that applicants in the industry [10, 11], but the fretting fatigue behavior on the AM60 magnesium alloy has not yet been investigated. A review of the literature about fatigue, fretting fatigue, and mechanical properties is discussed in the following phrases.

The effects of the thermomechanical process and subsequent heat-treating on the material behavior of a magnesium alloy (AM60) for the small fatigue crack growth have been examined by Chen et al. [12]. They looked at how crack growth rates varied over a variety of stress intensity variables in materials with various microstructures and values of the yield stress, applied under different processing circumstances.

The fatigue crack propagation process may have been a combination of transgranular and intergranular cracking, according to fracture surface characterization. In cast magnesium alloys, further investigations showed that the big casting pores were the most likely places for cracks to initiate [13, 14] whereas, both subsurface and surface fatigue crack initiations were observed, for wrought magnesium alloys [15]. It was founded that the mechanism of fatigue crack growth was related to the stress intensity factor [13, 16]. Others have shown that the sample exposure to water vapor dramatically enhanced the initiation and propagation of fatigue cracks [14, 17]. However, it has not been thoroughly defined how fatigue crack growth behaves for the cracks through the physically tiny crack regime under various processes and mechanical loads.

The hardness effects of the contact pad on the behavior of magnesium alloy (AZ61) under the fretting fatigue condition were investigated by Sadeler and Atasoy [18]. The findings demonstrated that the strength of fretting fatigue reduced with increasing hardness. The relative slip amplitude rose while the hardness increased and when the tangential amplitude was unaffected by the hardness. Olugbade et al. [19] presented a thorough evaluation of the electrochemical characteristics of magnesium alloys as biodegradable materials. Notably, corrosion fatigue and corrosion processes for surface-modified magnesium alloys have been thoroughly examined. Pîrvulescu et al. [20] discovered that increasing the test piece size affects fatigue lifetime due to a rise in the volume of defects in the material during an experimental investigation of the AM50A magnesium alloy.

Eisenmeier et al. [21] have investigated the cyclic behavior of AZ91 magnesium alloy at temperatures of 25 and 130˚C. Their fatigue test results were in good agreement with Manson-Coffin and Basquin models. They also illustrated that the cracks started from surface holes. Li et al. [22] investigated the influence of the strain amplitude on the fatigue behavior of AZ61 magnesium alloy. Their results showed that if the strain range was more than 0.5%, the shear crack would be significant. Rettberg et al. [23] presented the low-cycle fatigue behavior and the effect of heat treatment on magnesium alloys. The results depicted that in the amplitude of higher strains, the fatigue lifetime of AM60 alloy is higher due to its higher ductility.

Magnesium alloys are promising lightweight materials in various industrial fields such as automobiles, airplanes, trains, electronic components, and biomedical. Mechanical properties, corrosion, and fatigue have been reported [24, 25], but fretting fatigue has rarely been reported [26]. Consequently, comparing the pure and fretting fatigue of AM60 magnesium alloy could be considered as the novelty of this work. In other words, the experimental data of material characteristics under pure and fretting fatigue conditions in the AM60 magnesium alloy were presented with unusual trends, as a unique contribution to this research. After cyclic testing, field-emission scanning electron microscopy (FESEM) was utilized to examine the sample fracture behaviors, plus energy-dispersive spectroscopy (EDS) and X-Ray diffraction (XRD).

## 2. Materials and experiments

In the present work, the fretting fatigue characteristics of the AM60 magnesium alloy were investigated, compared to the pure fatigue lifetime through the high-cycle regime. It was discovered that the chemical composition was 5.9 wt.% Al, 0.4 wt.% Mn, 0.1 wt.% Zn, and the rest quantity was Mg. These values are comparable to AM60A based on the ASTM-B94 standard [10, 11].

The pure fatigue testing was carried out by means of the SFT-600 device, manufactured by Santam Company. With $R_\sigma$ = -1 (zero mean stress), a two-point bending (cantilever beam) rotary fatigue test apparatus was employed under fully-reversed loading conditions. The ISO-1143 standard contains test information. The loading frequency setting of 6000 rpm (100 Hz) is quite near to the working condition under rated-power for engines in automobiles [27, 28]. The fatigue lifetime testing limit was determined to be 1 million loading cycles [29]. Additionally, the reproducibility or repeatability of the experiment was evaluated using three standard samples (at least) under each value of the stress level. To establish the S-N curve, fatigue testing was conducted at five levels of stress amplitude, around 60, 80, 100, 120, and 140 MPa. Then, the stress amplitude was drawn versus the fatigue lifetime. Fig 1 illustrates the geometry of the standard sample.

To undertake fretting fatigue conditions, an additional device as the fretting force module was conceived and manufactured, allowing the fretting fatigue condition to be applied to the standard sample. Then, in the fretting module, the force bar was connected to the pads. Notably, two screws were used on the fatigue testing device that further fixed the module. The fretting module in the fatigue testing machine is depicted in Fig 2. Different springs of varying stiffnesses might be used in this apparatus to modify the amount of the contact force given to the specimen.

To replicate a comparable loading situation of fretting loading on the studied material, the fretting pads were cut from the piston rings, with the material of ductile cast iron in this investigation. For such an investigation, the FESEM image (Mira3 TESCAN, SEM HV: 15.0 kV) was utilized for the fractography. The objective was to find the damage mechanism from the fracture surface of samples. Moreover, the XRD analysis was also used to determine the phase in the microstructure.

## 3. Results and discussion

### 3.1. Microstructure

Fig 3 depicts the XRD pattern for the materials as received (AM60). Aside from the Mg element, an intermetallic phase of Al-Mn ($Al_6Mn$) and Mg-Al ($Mg_{17}Al_{12}$) was detected on the microstructure. These results could be confirmed by the literature [30].

The results of scanning electron microscopy (SEM) images with the energy-dispersive spectroscopy (EDS) results (provided in the previous works [10, 11]) can be respectively seen in Fig 4 and Table 1. Three phases, including Mg (α phase), $Mg_{17}Al_{12}$ (β phase), and $Al_6Mn$ detected in the XRD pattern.

Another reference [31] confirmed this microstructure. Additionally, the MgO phase can be detected in the microstructure, as depicted in Table 1 and Fig 4. It indicates that there was surface oxidation. The alloy exhibited a dendritic microstructure and second-phase particles that were dispersed in the interdendritic regimes, as could be shown in the literature [11, 31]. The alloy exhibited bulky $Mg_{17}Al_{12}$ particles and discontinuous precipitations, according to the SEM picture [11, 31]. The majority of the β-$Mg_{17}Al_{12}$ intermetallic particles found in interdendritic zones were bulky and irregular in shape. The remainder of the material was discovered as discontinuous β-$Mg_{17}Al_{12}$ precipitates [11, 31].

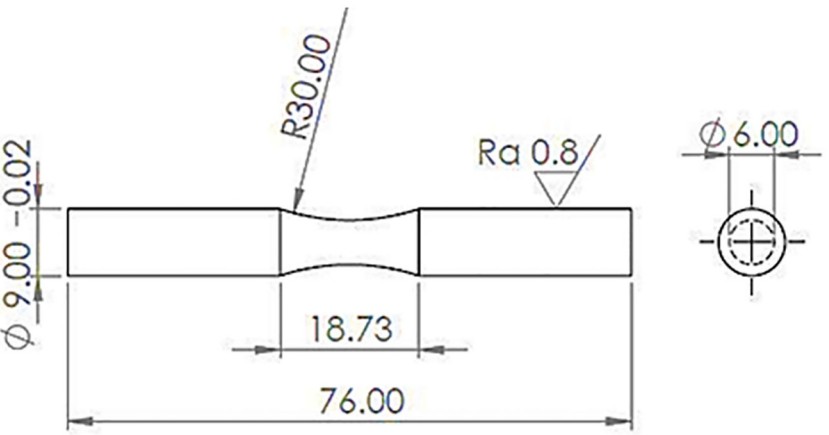

**Fig 1. The standard sample dimension (millimeters) for pure and fretting fatigue testing.**

## 3.2. Pure and fretting fatigue lifetime

Fig 5 compares the S-N (stress-lifetime) curves of the AM60 magnesium alloy under pure fatigue (PF) and fretting fatigue (FF) conditions ((a): for all data, (b): for average data). It is worth noting that at least, three tests were done at each stress level. The fatigue lifetimes of these experiments were averaged to find the mean value. Therefore, in one case (Fig 5(a)), all experimental data are presented and, in another case, only average lifetimes are reported to find a specific trend of the amazing Epsilon-shaped behavior. As another note, to compare pure fatigue and fretting fatigue behaviors, the bending stress was drawn versus the fatigue lifetime. Under fretting fatigue conditions, the real stress of samples is a combination of the bending stress and the contact stress. This value could be calculated by finite element simulations. However, the bending stress is similar in both pure fatigue and fretting fatigue, which is used for the comparison of pure fatigue and fretting fatigue data.

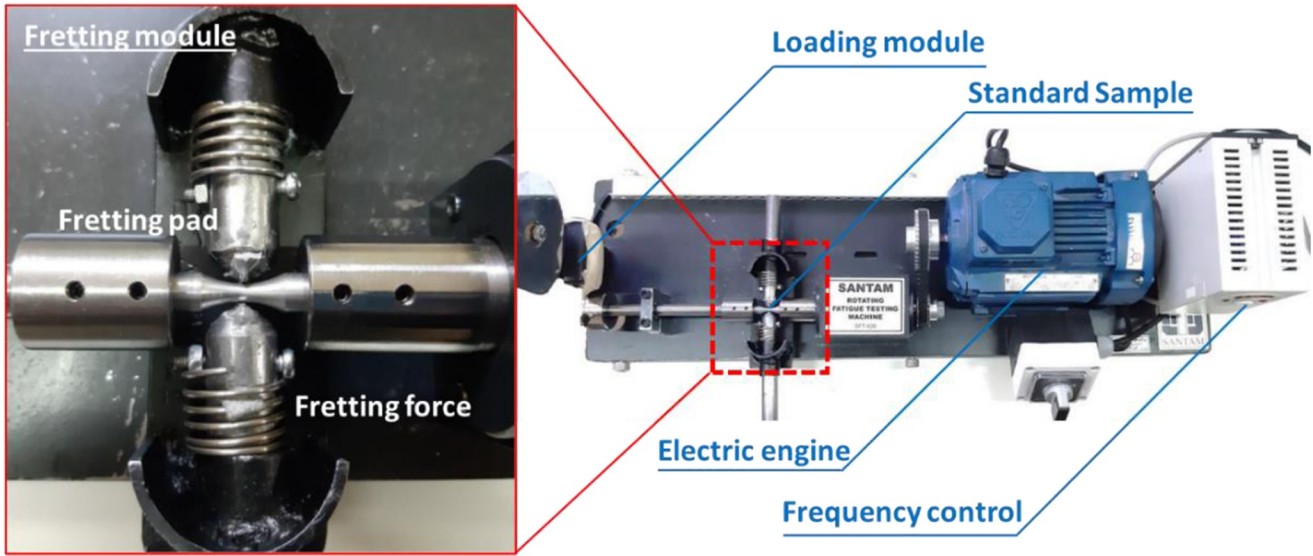

**Fig 2. The rotary bending fatigue testing machine with the fretting module.**

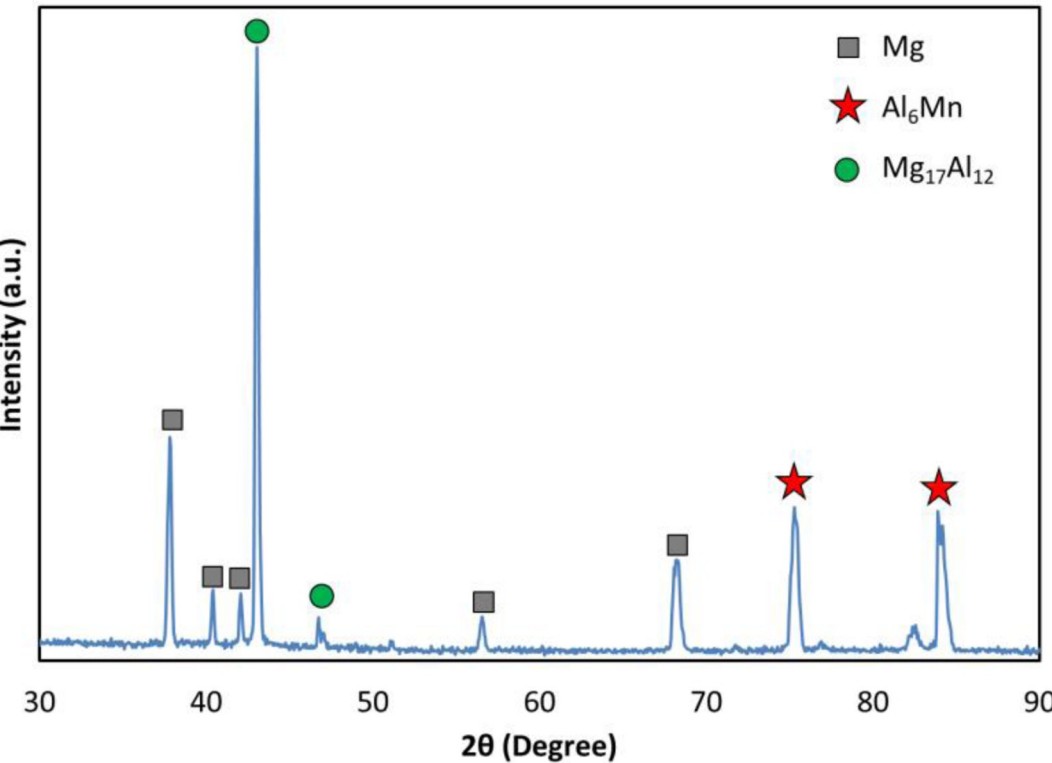

**Fig 3. The XRD pattern for the AM60 magnesium alloy.**

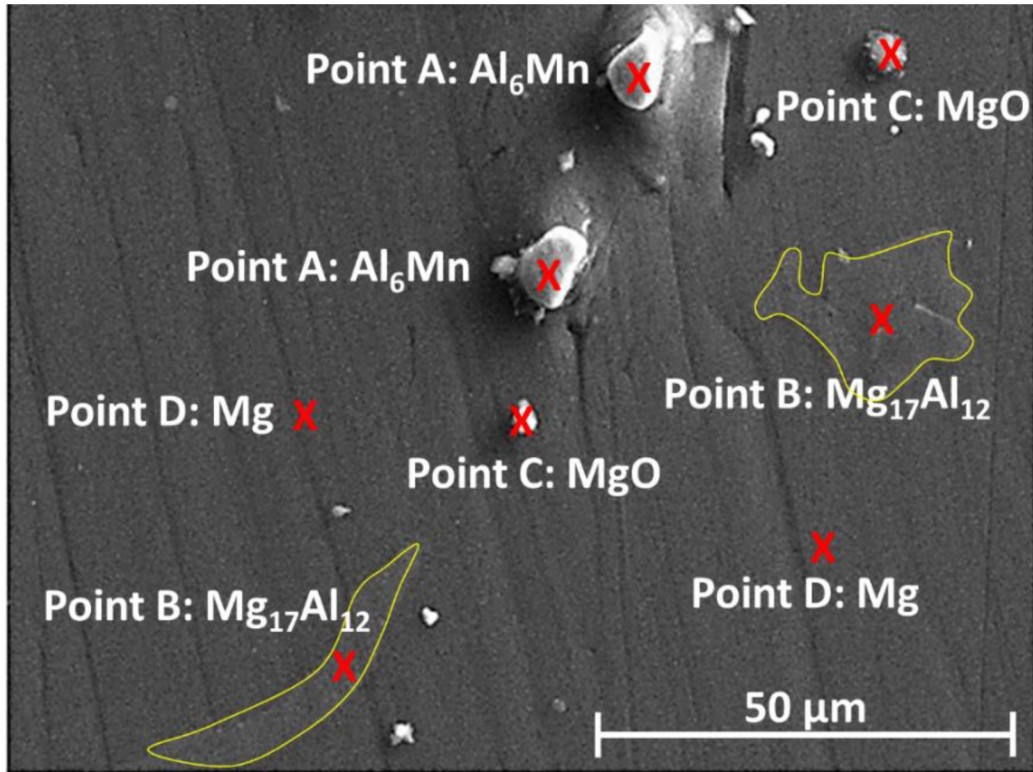

**Fig 4. The SEM image of the AM60 magnesium alloy.**

**Table 1. The EDS result in Fig 4 for the elemental analysis in the AM60 magnesium alloy.**

| Points in Fig 4 | Elements | Weight percent (%) | Atomic percent (%) |
|---|---|---|---|
| Point A: $Al_6Mn$ | Mg | 1.65 | 2.58 |
| | Al | 38.75 | 54.76 |
| | Si | 1.95 | 2.65 |
| | Mn | 57.65 | 40.01 |
| Point B: $Mg_{17}Al_{12}$ | Mg | 62.24 | 64.84 |
| | Al | 37.24 | 34.95 |
| | Cu | 0.52 | 0.21 |
| Point C: MgO | Mg | 68.88 | 60.86 |
| | Al | 2.92 | 2.33 |
| | Ca | 0.47 | 0.25 |
| | Ni | 0.68 | 0.25 |
| | O | 27.05 | 36.32 |
| Point D: Mg | Mg | 96.09 | 96.47 |
| | Al | 3.91 | 3.53 |

The results in the state of pure fatigue were almost similar to the study by Chen et al. [12]. According to the data shown in Fig 5, the fretting fatigue condition had a shorter fatigue lifetime than the pure fatigue condition. At the highest level of stress (120 MPa) and the lowest level of stress (80 MPa), respectively, the fatigue lifetime under the fretting fatigue condition decreased by 44.8% and 91.0% in comparison to the pure fatigue condition. Notably, the highest and lowest stress amounts examined in the comparison were 120 MPa and 80 MPa, respectively. Using the average values in Fig 5(b), these values were derived. These statements imply that the reduction in fretting fatigue lifetime versus pure fatigue lifetime was greater through the high-cycle fatigue regime (at the lowest level of stress) than through the low-cycle fatigue regime (at the highest level of stress). Moreover, in Fig 5, curve fitting was utilized to identify the material properties. It is possible to represent the link between the fatigue lifetime and the stress amplitude, based on the following equation [29]:

$$\sigma_a = \sigma'_f \left(2N_f\right)^b \tag{1}$$

The fatigue lifetime is denoted by $N_f$, the stress amplitude is depicted by $\sigma_a$, and the fatigue strength coefficient is mentioned by $\sigma'_f$. Additionally, $b$ also stands for the exponent of fatigue strength.

The coefficient of determination ($R^2$) for all experimental data for the pure fatigue and fretting fatigue conditions was then determined using the regression analysis in Fig 5, and it was found to be 94.07% and 68.34%, respectively. The $R^2$ value was 95.34% and 80.51% for average values of pure fatigue and fretting fatigue. Some researchers considered a value higher than 90% for $R^2$ to be acceptable for pure fatigue conditions [32]. Table 2 also shows the high-cycle fatigue characteristics in the AM60 magnesium alloy under pure fatigue and fretting fatigue. As depicted in Table 2, the fretting fatigue condition enhanced the fatigue strength but with a higher negative curve slope ($b$). In fretting fatigue loading versus pure fatigue, the fatigue strength coefficient ($\sigma'_f$) improved significantly (around 39%). When fretting fatigue was used instead of pure fatigue, The fatigue strength exponent ($b$) increased by 38% in absolute value.

The fretting fatigue S-N diagram had an unusual form, as shown in Fig 5(b). In a good agreement trend, the stress-lifetime curve for aluminum alloys exhibited an epsilon-shaped

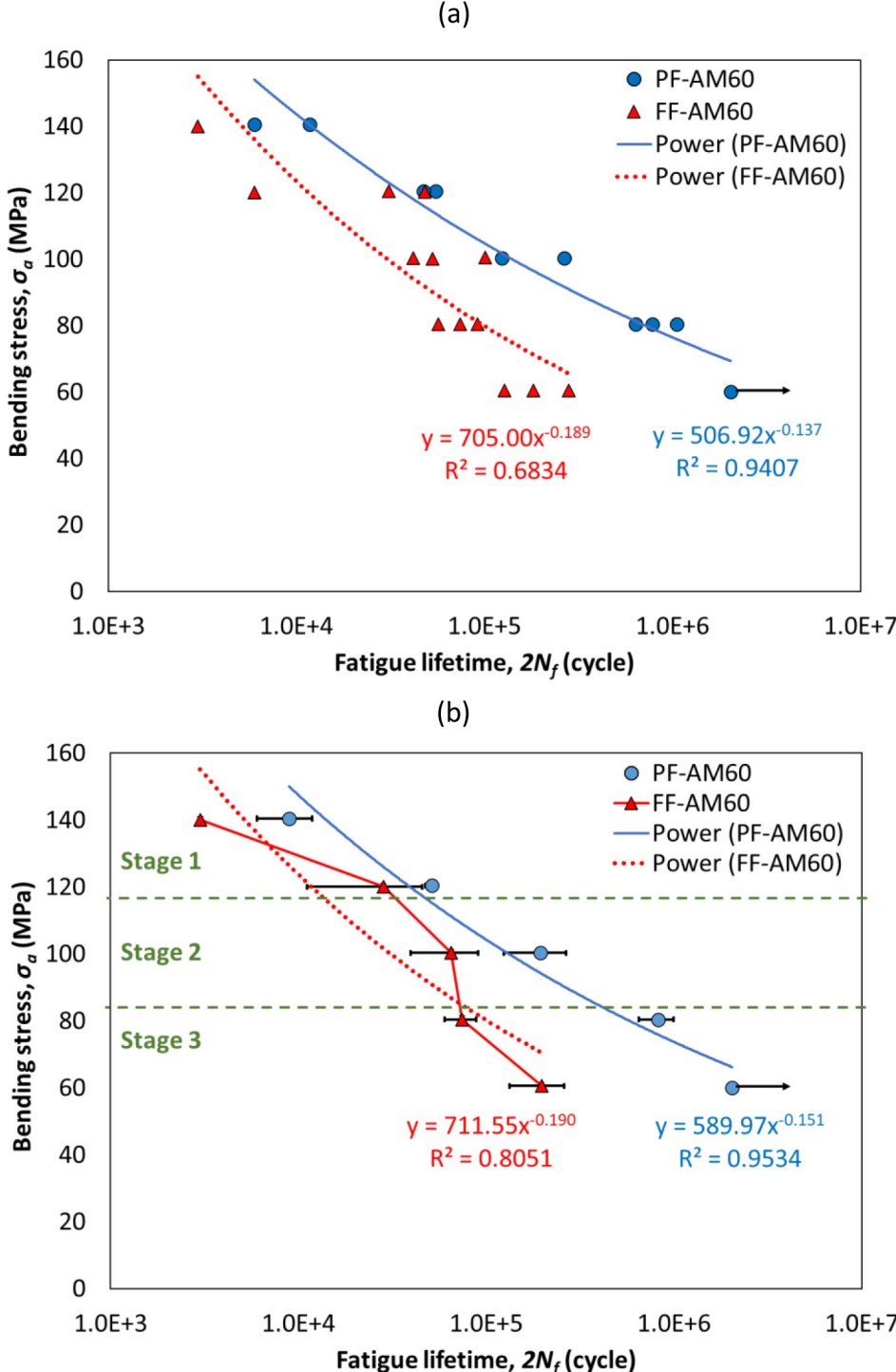

**Fig 5. The data of pure fatigue (PF) and fretting fatigue (FF) for the bending stress versus the fatigue lifetime: (a) all raw data and (b) average values.**

**Table 2. Fatigue properties of the studied AM60 magnesium alloy.**

| Condition | Using all experimental data | | Using average values | |
|---|---|---|---|---|
| | $\sigma'_f$ (MPa) | $b$ (-) | $\sigma'_f$ (MPa) | $b$ (-) |
| Pure fatigue | 506.92 | -0.137 | 589.97 | -0.151 |
| Fretting fatigue | 705.00 | -0.189 | 711.55 | -0.190 |

behavior under bending fretting fatigue conditions, according to Peng et al. [33] and Parast and Azadi [34]. It was found that raising the level of stress reduced the fretting fatigue lifetime, unless the stress level was less than 120 MPa. In other words, raising the stress level enhanced the fretting fatigue lifetime. Peng et al. [33, 35, 36] indicated that similar to the fretting fatigue phenomena, three forms of fretting fatigue under bending loads may be seen. By raising the stress level, these categories could be split into three regions or stages. These regimes were the slip regime (SR), which was represented by "Stage 1" in Fig 5(b), the mixed fretting regime (MFR), which was represented by "Stage 2" in Fig 5(b), and the partial slip regime (PSR), which was represented by "Stage 3" in Fig 5(b). Despite the fact that the crack first appeared at the maximum wear rate in the slip regime (SR), The micro-crack was eliminated during the nucleation phase and was unable to spread. As a result, in an unexpected trend, the fretting fatigue lifetime rose slightly [34, 36].

### 3.3. Fracture analysis

The failure behavior of AM60 magnesium alloys was evaluated using FESEM on the fracture surface of samples. All evaluated surfaces were subjected to identical fatigue test conditions, with stress levels of 80 and 120 MPa (Fig 6(a) and 6(c): 80 MPa, Fig 6(b) and 6(d): 120 MPa).

As an issue according to Fig 6(a), a small scratched area can be seen, which was due to wear between two surfaces of the sample (at the side of fixed and loading parts) after separation at the final cycles of fatigue testing.

Based on the fractography analysis of the hexagonally close-packed (HCP) metal, faceted morphologies have frequently been seen following fatigue failure, particularly in titanium alloys [37] and magnesium alloys [15, 38]. These features from the fracture morphology could result from cleavage and quasi-cleavage marks along the proper crystal planes.

As another important note in Fig 6(c) and 6(d), the fretting area can be seen. This region was so larger in the sample under 80 MPa of the stress level. However, the fretting area was so small in the specimen under 120 MPa, which the fatigue phenomenon was a dominant failure than the fretting fatigue. This issue could be also claimed from the fatigue lifetime. Under 120 MPa of the stress level, the lifetime was similar under pure fatigue and fretting fatigue conditions, as $51,300\pm3,900$ and $28,333\pm17,250$ cycles, respectively (with a ratio of 1.8). This difference between pure fatigue and fretting fatigue lifetimes was larger as $824,333\pm172,980$ and $74,400\pm14,373$ cycles, respectively (with a ratio of 11.1), under 80 MPa of the stress level.

Figs 7–10 depict flat cleavage/quasi-cleavage marks including some patterns of striation (on the micro-scale compare to beach marks on the macro-scale) and also micro-cracks and their rough growth paths, which were randomly oriented serrated surfaces on the sample fracture surface. A high-cycle fatigue analysis of a cast AM60 magnesium alloy revealed similar fracture surface characteristics [13]. Transgranular crack growth via dendrite cells was linked with flat planes (cleavage) with striation marks, while intergranular crack growth through the β-$Mg_{17}Al_{12}$ phase was correlated with rough areas with serrated surfaces [13]. The brightness of the specimen fracture surfaces also changed when the former crack development mechanism

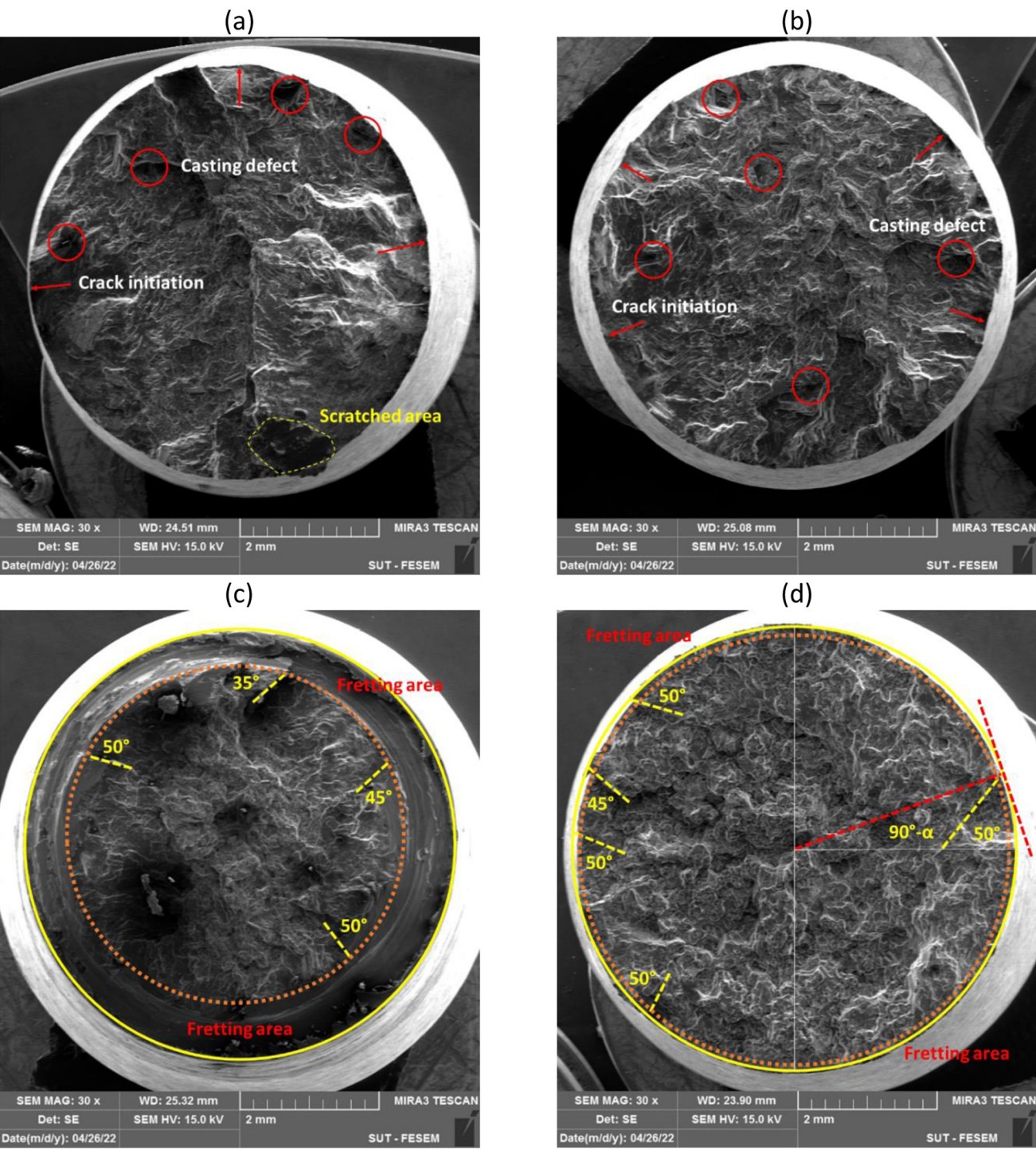

**Fig 6. The fracture surface under pure fatigue at the stress level of (a) 80 MPa and (b) 120 MPa and under fretting fatigue at the stress levels of (c) 80 MPa and (d) 120 MPa.**

gave way to the latter with a rise in the stress intensity component, which was the driving force behind crack growth. The driving force should be strong enough to produce a premature fracture or to cause β-phase debonding [13, 39]. The majority of the facets in the specimen fracture surface were intergranular, as shown by Hashemi et al. [39]. The rate of crack propagation is

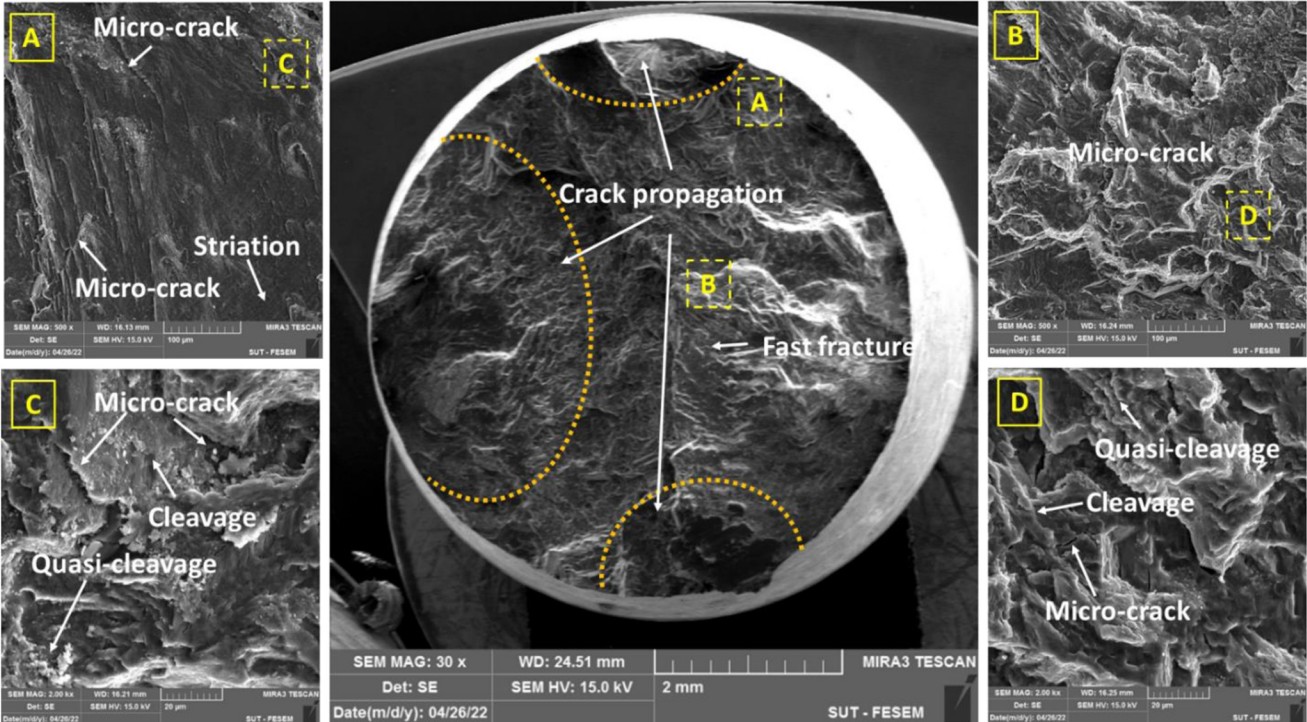

**Fig 7. The fracture surface under pure fatigue conditions (the stress level: 80 MPa).**

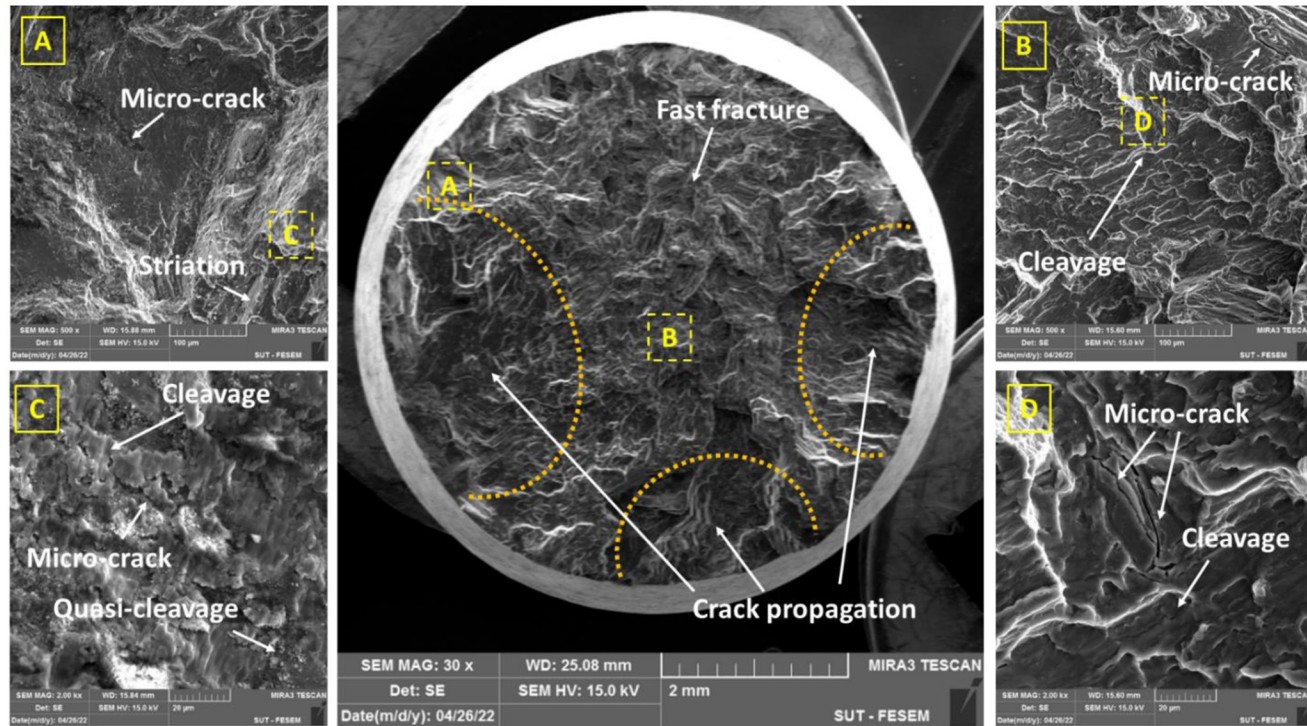

**Fig 8. The fracture surface under pure fatigue conditions (the stress level: 120 MPa).**

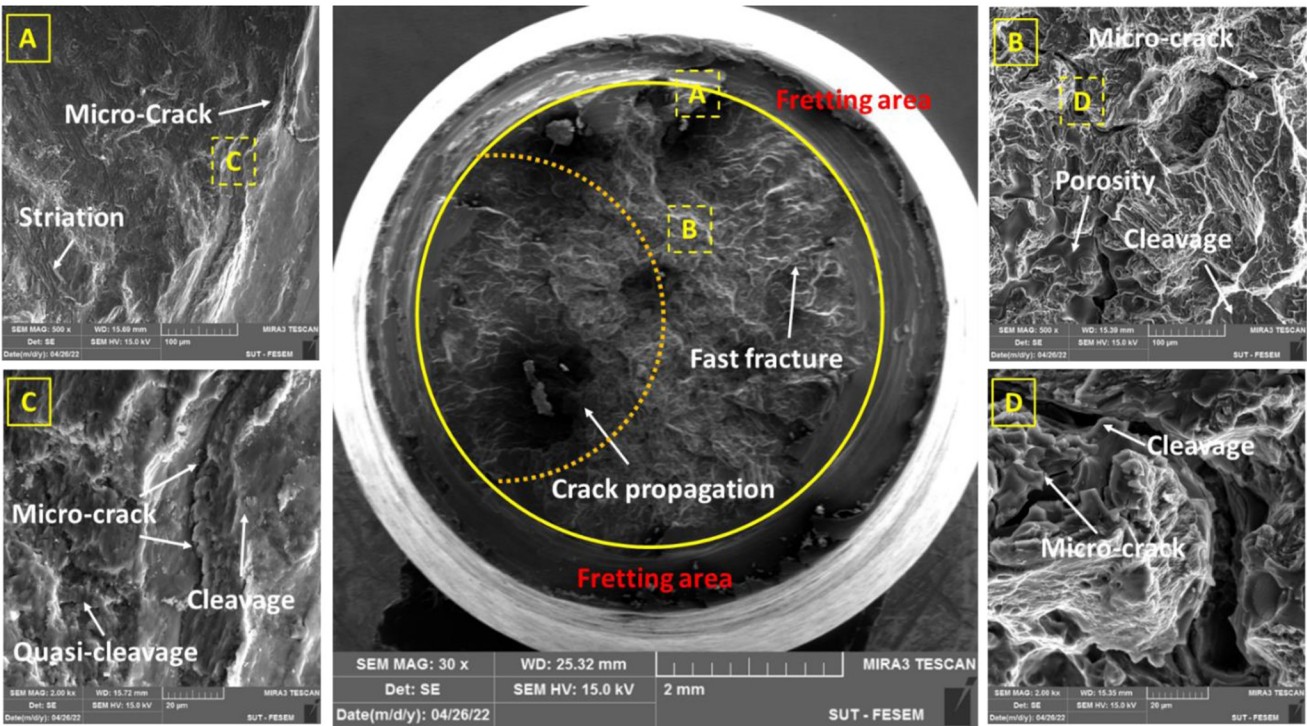

**Fig 9. The fracture surface under fretting fatigue conditions (the stress level: 80 MPa).**

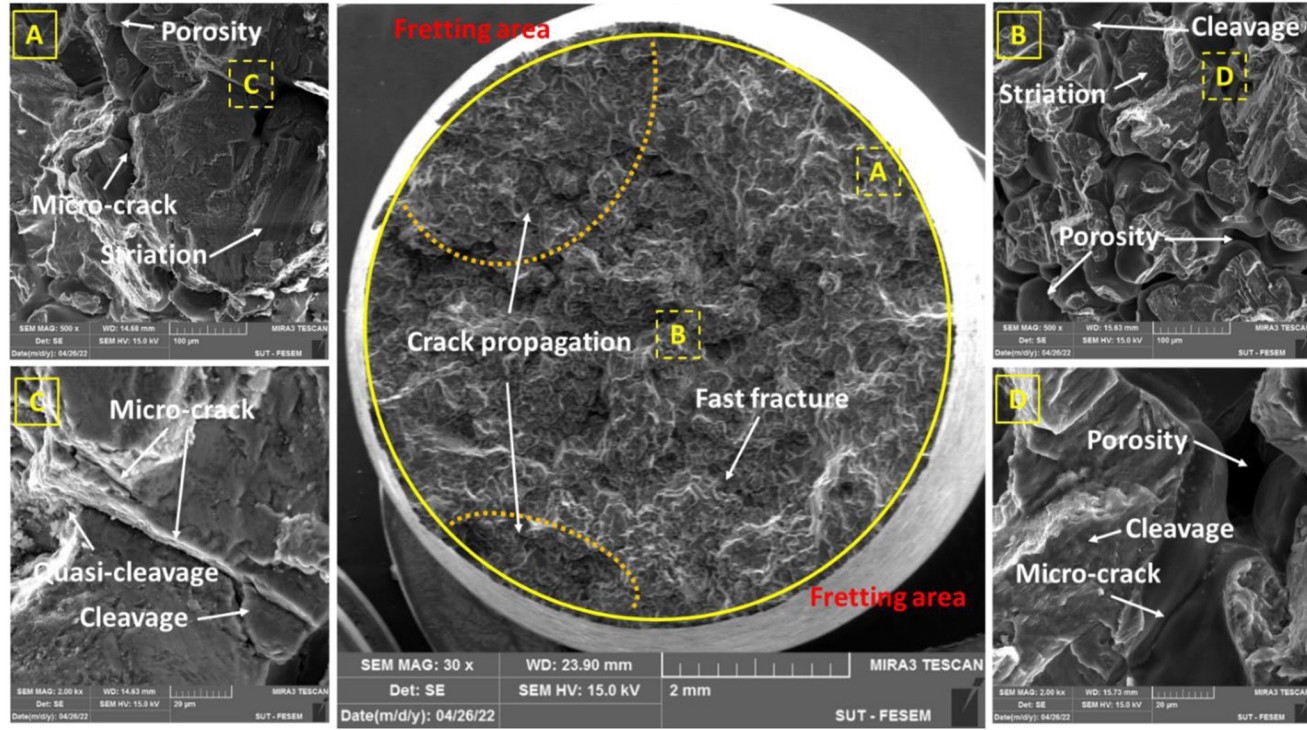

**Fig 10. The fracture surface under fretting fatigue conditions (the stress level: 120 MPa).**

considerably decreased by intergranular facets. This impact can be explained by the increased tortuosity that is produced in the crack passage and the crack divergence from the maximum stress plain.

As it can be seen in Figs 9 and 10, wear, and actually, the effect of fretting is evident in the area of the contact edge with the pads. This type of failure surface for fretting fatigue conditions was also observed in Sadeler and Atasoy [18]. As observed in Fig 6(c) and 6(d), the fretting fatigue crack propagation began at 35–50 degrees, respectively. According to the results of Peng et al. [35, 40], the fretting fatigue crack propagation began at a 30-degree angle toward the sample surface.

Because the loading circumstances in this investigation were bending, the most stress is induced on the sample surface. Then, as a result, the majority of fatigue cracks begin close to the surface. Of course, this result was indicated by Nascimento et al. [41] and Oliya et al. [29]. Additionally, fretting fatigue cracks are also produced in this region because the contact stresses have their largest local values near the contact surface due to stress concentration. As a result, it appears that the shorter fretting fatigue lifetime is caused by a shorter fracture formation lifetime and a faster rate of crack propagation after the presence of stress concentration under wear fatigue loading conditions [34]. One of the main mechanisms of crack growth in wear fatigue is the wedge effect [34]. This issue happens when the shavings from wear enter the small cracks of wear fatigue and increase the speed of the crack growth [34]. Fig 6(c) makes the consequences of wear damage on the sample edge quite evident.

In Figs 7–10, in some cases, the porosity due to casting defects could be also seen in FESEM images, based on their known shape. They often take the shape of a circle in cross-section but can get the form of an irregular linear crack. This is a mechanism where a volumetric contraction of a metal solidifies and the jagged edges form, when there is not enough liquid to fill the shrinkage. These defects were due to solidification shrinkage during casting that will decrease the fatigue lifetime and could be the region for the crack initiation. According to the work of other researchers in the majority of the instances, casting defects were found to be the cause for the initiation of fatigue cracks and then, sample failures [21, 42–44].

Table 3 shows the number and length of microcracks in pure fatigue and fretting fatigue conditions. According to the findings in Table 3, the average crack length was greater at higher stress levels than at lower stress levels in both test conditions (fretting fatigue and pure fatigue). As a confirmation, this issue was also reported by Rezanezhad et al. [27] for aluminum alloys. Higher densities of cracks were observed under the highest stress level or the low-cycle fatigue regime, compared to the lowest stress level or the high-cycle fatigue regime. Furthermore, the number of cracks (at high- and low-stress levels) was found to be lower in the fretting fatigue conditions than in the pure bending fatigue conditions; however, the average crack length was 212.82% and 259.47% greater in the fretting fatigue situation, respectively, than in the pure fatigue condition. Considering that the number of cracks in the pure fatigue condition was more than one in the fretting fatigue; however, the crack length was longer in the fretting fatigue condition. Then, the fatigue lifetime of the samples in the

**Table 3. The micro-crack characteristics on 2000X-FESEM images (in crack propagation regions).**

| Stress level | Pure fatigue | | Fretting fatigue | |
|---|---|---|---|---|
| | Number | Length (μm) | Number | Length (μm) |
| 80 MPa | 45 | 4.96±2.95 | 15 | 17.83±22.36 |
| 120 MPa | 39 | 7.41±4.20 | 11 | 23.18±17.72 |

fretting condition is lower than in the pure condition. It can be concluded that the crack length was more significant than the number of cracks in the fatigue lifetime. In general, both cleavage and quasi-cleavage features were seen and consequently, the fracture behavior for all samples was brittle.

In addition, the EDS map analysis was done from the fracture surfaces of the material. These images can be seen in Fig 11. In this figure, it is clear that the magnesium phase and intermetallic phases were present on the fracture surface of all samples. In Fig 11, orange, purple, and blue colors are Mg, Al, and Mn elements, respectively.

According to the results of Horstemeyer et al. [13], as well as Fig 11, these intermetallic phases are susceptible to the initiation of microcracks. Therefore, the microcracks are formed near intermetallic deposits, especially from Mg-Al firstly (more significant) and then, Al-Mn intermetallics, and lead to final failure. Based on Fig 11, the microcracks initiated from the region of the $Mg_{17}Al_{12}$ intermetallic phase in the magnesium matrix. Mokhtarishirazabad et al. [45] illustrated that the $Mg_{17}Al_{12}$ phase caused lower strength, especially at high temperatures, where the creep phenomenon occurred. As a confirmation of the presented results with the literature [13, 45], the "X" area, which showed the crack path in Fig 11, contained the Al or Mn elements as intermetallic phases. Most cracks were from Mg-Al intermetallic phases and one case was due to Mn. Notably, the Mg matrix alone was not the origin of cracks, based on its own effects, since it is softer than intermetallic phases.

## 4. Conclusions

In this article, the obtained outcomes can be summarized on pure bending fatigue and also fretting fatigue characteristics of the AM60 magnesium alloy, as follows,

- At the maximum level of stress (120 MPa through low-cycle fatigue regime) and the lowest level of stress (80 MPa through high-cycle fatigue regime), respectively, the fretting fatigue lifetime was reduced by 44.8% and 91.0% compared to the pure fatigue condition.

- The number of material characteristics, such as the fatigue strength exponent and coefficient, increased under the fretting fatigue condition compared to pure fatigue for both the average values and all experimental data. In comparison to pure fatigue, fretting fatigue loading was significantly improved (by roughly 39%) by the fatigue strength coefficient ($\sigma'_f$). When compared to pure fatigue, the fatigue strength exponent ($b$) increased by 38% in an absolute value.

- The fretting fatigue S-N diagram had an unusual amazing trend (epsilon-shaped); it was seen that while the stress amplitude was enhanced, the lifetime of fretting fatigue decreased unless the stress level was below 120 MPa.

- Due to quasi-cleavage and cleavage features on the fracture surface of samples, the fracture behavior of the AM60 magnesium alloy was brittle.

- The average crack length was longer at greater stress levels than at lower stress levels in both condition tests (fretting fatigue and pure fatigue). The average crack length in fretting fatigue conditions in high-stress and low-stress levels was 212.82% and 259.47% higher than the average crack length in pure fatigue conditions, respectively. In addition, it was observed that the number of cracks (in high and low-stress levels) was lower in fretting fatigue conditions than in pure fatigue conditions.

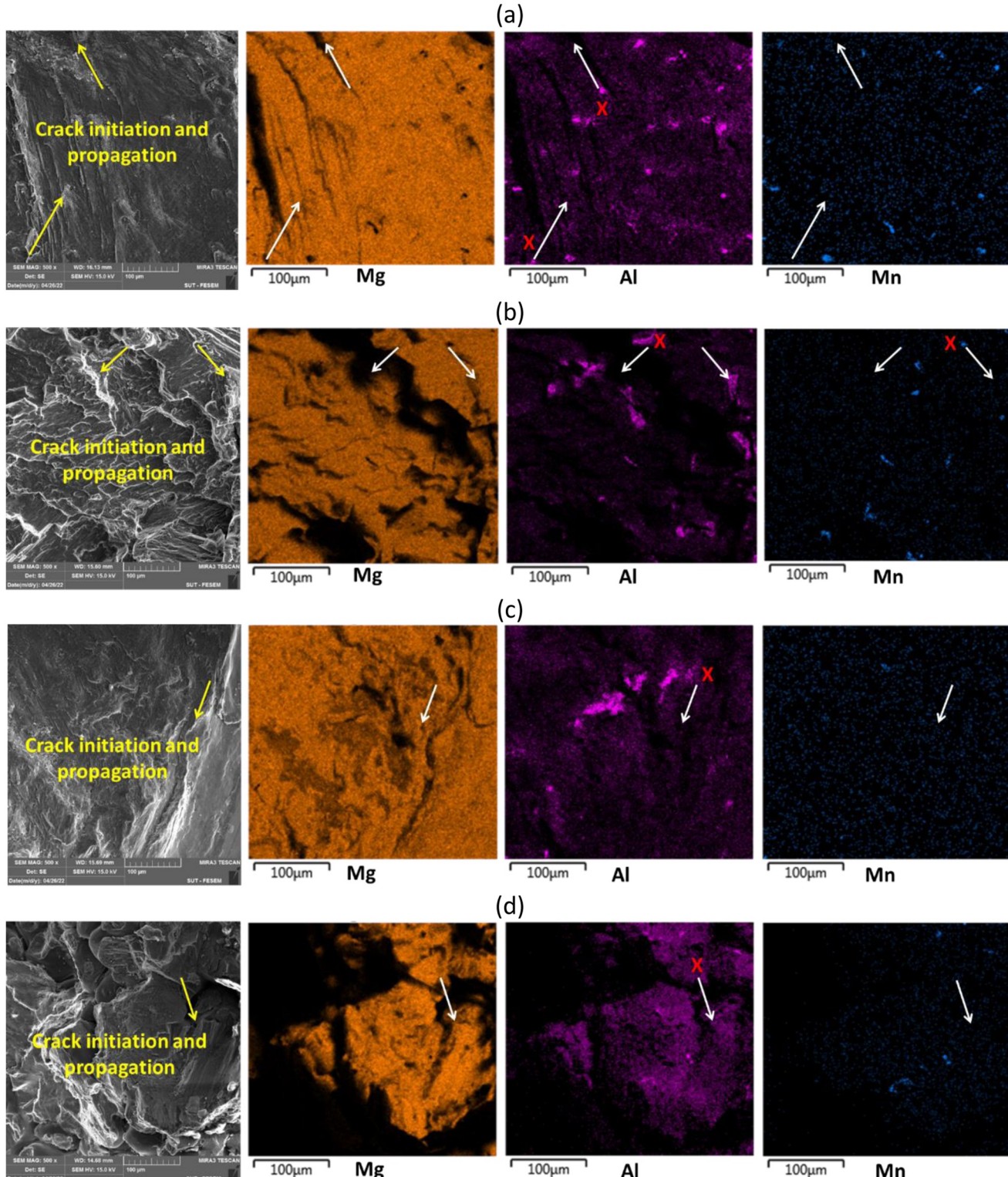

**Fig 11. FESEM images with the EDS analysis of sample fracture surfaces for (a) pure fatigue at 80 MPa and (b) 120 MPa of the stress level, (c) fretting fatigue at 80 MPa) and (d) 120 MPa of the stress level (Notes: The arrows start with the crack initiation and their directions show the crack propagation paths.** Moreover, the symbol "X" depicts the concentration of the related Al or Mn intermetallic phases in the Mg matrix near the region of the crack initiation).

## Author Contributions

**Conceptualization:** Mohammad Azadi.

**Data curation:** Saeid Rezanezhad.

**Formal analysis:** Saeid Rezanezhad.

**Funding acquisition:** Mohammad Azadi.

**Investigation:** Saeid Rezanezhad, Mohammad Azadi.

**Methodology:** Saeid Rezanezhad, Mohammad Azadi.

**Project administration:** Mohammad Azadi.

**Resources:** Mohammad Azadi.

**Software:** Saeid Rezanezhad.

**Supervision:** Mohammad Azadi.

**Validation:** Mohammad Azadi.

**Visualization:** Saeid Rezanezhad, Mohammad Azadi.

**Writing – original draft:** Saeid Rezanezhad.

**Writing – review & editing:** Mohammad Azadi.

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
