## [Decision Letter · Decision Letter 0]

2 Jan 2023

PONE-D-22-31978Amazing Epsilon-shaped trend for fretting fatigue characteristics in AM60 magnesium alloy under stress-controlled cyclic conditions at bending loads with zero mean stressPLOS ONE

Dear Dr. Azadi,

Thank you for submitting your manuscript to PLOS ONE. After careful consideration, we feel that it has merit but does not fully meet PLOS ONE’s publication criteria as it currently stands. Therefore, we invite you to submit a revised version of the manuscript that addresses the points raised during the review process.Please revise your manuscript according to the reviewers comments.

We look forward to receiving your revised manuscript.

Kind regards,

Khalil Abdelrazek Khalil, Ph.D.

Academic Editor

PLOS ONE

Journal Requirements:

"No"

Additional Editor Comments:

Please take care of the reviewer comments especially the typo in the manuscript.

Reviewers' comments:

Reviewer's Responses to Questions

**Comments to the Author**

1. Is the manuscript technically sound, and do the data support the conclusions?

Reviewer #1: Yes

Reviewer #2: Yes

2. Has the statistical analysis been performed appropriately and rigorously? 

Reviewer #1: Yes

Reviewer #2: Yes

3. Have the authors made all data underlying the findings in their manuscript fully available?

Reviewer #1: Yes

Reviewer #2: Yes

4. Is the manuscript presented in an intelligible fashion and written in standard English?

Reviewer #1: Yes

Reviewer #2: Yes

5. Review Comments to the Author

Reviewer #1: 1. The second paragraph in the Introduction section is too lengthy.

2. In the first paragraph of Section 3.2, the authors mentioned the averaged values in Figure 5(b) and the values were derived. The authors are invited to explain more about this statement as it is unclear.

3. The manuscript contains some typos.

Reviewer #2: Review comments:-

1. What type of rotary fatigue testing device can be used in these experiments?

2. Mention the importance of field-emission scanning electron microscopy (FESEM)?

3. how the fatigue lifetime compared with the stress values of pure fatigue (PF) and fretting fatigue?

4. Casting defect identified using which instrument? Mention the specifications of the testing device.

5. If the stress level will be more means, Did the crack propagations will become more or less?

6. In table 3, whether the length has been increased due to the increase of stress level? Or the Mg contribution of materials composition?

7. Why were the different color SEM images displayed in figure 11? Any specific reason beyond this?

6. PLOS authors have the option to publish the peer review history of their article (what does this mean?). If published, this will include your full peer review and any attached files.

Reviewer #1: No

Reviewer #2: No

---

## [Author Response · Author response to Decision Letter 0]

9 Jan 2023

Our article, entitled “Amazing Epsilon-shaped trend for fretting fatigue characteristics in AM60 magnesium alloy under stress-controlled cyclic conditions at bending loads with zero mean stress”, is reviewed by respected reviewers. First of all, the authors should thank you for this review. Then, the authors have tried their bests to address all comments. The changes were highlighted in yellow-colored sentences. Moreover, the answers to each comment are provided in the following paragraphs. 

Regards,

M. Azadi, PhD.

Faculty of Mechanical Engineering, Semnan University, Semnan, Iran

Additional Editor Comments:

Please take care of the reviewer comments especially the typo in the manuscript.

Answer: Thank you for the chance for revising the article. The authors have tried their bests to address all comments in the revised article. Moreover, the whole manuscript is read again to find any mistakes. In addition, the whole text is checked by the Grammarly software with the overall score of 87. 

Reviewer #1: 

1. The second paragraph in the Introduction section is too lengthy.

Answer: The authors should thank the respected reviewer for his/her nice comments. Then, all comments were completely addressed in the revised article. 

For this comment, the second paragraph in the introduction is separated into four paragraphs to prevent the lengthy issue. Moreover, some sentences were eliminated to shorten this part. 

2. In the first paragraph of Section 3.2, the authors mentioned the averaged values in Figure 5(b) and the values were derived. The authors are invited to explain more about this statement as it is unclear.

Answer: Thank you for the nice suggestion. The authors have added the following text for a better explanation in the revised article, as follows,

It is worth noting that at least, three tests were done at each stress level. The fatigue lifetimes of these experiments were averaged to find the mean value. Therefore, in one case (Figure 5(a)), all experimental data are presented and, in another case, only average lifetimes are reported to find a specific trend of the amazing Epsilon-shaped behavior. 

3. The manuscript contains some typos.

Answer: Sorry for this mistake. The whole manuscript is read again to find any mistakes. Moreover, the whole text is checked through the Grammarly software and the overall score of 87 is obtained. 

Reviewer #2: 

1. What type of rotary fatigue testing device can be used in these experiments?

Answer: The authors should thank the respected reviewer for his/her nice comments. Then, all comments were completely addressed in the revised article. 

For this comment, the cantilever beam type was used with two-point bending conditions. This issue is highlighted in the text, as follows,

The pure fatigue testing was carried out by means of the SFT-600 device, manufactured by Santam Company. With Rσ = -1 (zero mean stress), a two-point bending (cantilever beam) rotary fatigue test apparatus was employed under fully-reversed loading conditions.

2. Mention the importance of field-emission scanning electron microscopy (FESEM)?

Answer: The objective was to find the damage mechanisms. This issue is added to the revised article, as follows, 

For such an investigation, the FESEM image (Mira3 TESCAN, SEM HV: 15.0 kV) was utilized for the fractography. The objective was to find the damage mechanism from the fracture surface of samples. Moreover, the XRD analysis was also used to determine the phase in the microstructure.

3. how the fatigue lifetime compared with the stress values of pure fatigue (PF) and fretting fatigue?

Answer: The respected reviewer is correct. However, to find the real stress value, the finite element simulation needs to be done. Moreover, this type of presentation is common in the literature, such as Chen et al. [12], Peng et al. [33], and Parast and Azadi [34].

In order to address this nice comment, the real stress in each fretting fatigue samples could be a combination of the bending stress and the contact stress. The bending stress is similar in both pure fatigue and fretting fatigue. Therefore, on the S-N curves, the title of the vertical axis is changed from “stress” to “bending stress”, in Figure 5, besides mentioning in the figure title. Moreover, the following description is also added to the revised article, as follows, 

As another note, to compare pure fatigue and fretting fatigue behaviors, the bending stress was drawn versus the fatigue lifetime. Under fretting fatigue conditions, the real stress of samples is a combination of the bending stress and the contact stress. This value could be calculated by finite element simulations. However, the bending stress is similar in both pure fatigue and fretting fatigue, which is used for the comparison of pure fatigue and fretting fatigue data. 

4. Casting defect identified using which instrument? Mention the specifications of the testing device.

Answer: The porosity and the casting defects were found by the FESEM images, based on their known shapes. This issue is mentioned in the revised article, as follows,

In Figures 7-10, in some cases, the porosity due to casting defects could be also seen in FESEM images, based on their known shape. They often take the shape of a circle in cross-section but can get the form of an irregular linear crack. This is a mechanism where a volumetric contraction of a metal solidifies and the jagged edges form, when there is not enough liquid to fill the shrinkage. These defects were due to solidification shrinkage during casting that will decrease the fatigue lifetime and could be the region for the crack initiation. 

Moreover, the used FESEM model was Mira3 TESCAN, which is also mentioned in the revised article, as follows,

For such an investigation, the FESEM image (Mira3 TESCAN, SEM HV: 15.0 kV) was utilized for the fractography. The objective was to find the damage mechanism from the fracture surface of samples. Moreover, the XRD analysis was also used to determine the phase in the microstructure.

5. If the stress level will be more means, Did the crack propagations will become more or less?

Answer: As known, when the stress level is high, the cycles to initiate the crack are short and the cycles for crack propagation are long. In other words, cracks will occur very soon and then, the whole lifetime includes a large portion of crack propagation. This issue will be reversed when the stress is low, through the high-cycle fatigue regimes. 

These descriptions were also mentioned in the book of “Metal Fatigue in Engineering” generally for metals and also by Rezanezhad et al. [27] (based on the following image) for aluminum alloys.

An image from the reference: Rezanezhad et al. [27]

In order to address this comment, the following text is added to the revised article, as follows,

As a confirmation, this issue was also reported by Rezanezhad et al. [27] for aluminum alloys. Higher densities of cracks were observed under the highest stress level or the low-cycle fatigue regime, compared to the lowest stress level or the high-cycle fatigue regime.

6. In table 3, whether the length has been increased due to the increase of stress level? Or the Mg contribution of materials composition?

Answer: As mentioned in the previous comment, when the stress increased, the crack length increased too. However, the material composition was similar in all samples. 

If the respected reviewer means from the EDS analysis, since these images have different magnification (500X) from FESEM images for Table 3 (2000X), they could not be used for this conclusion. Moreover, the failure mechanism is from intermetallic and not from the soft Mg matrix. The “X” area in Figure 11 is considered for the location the crack, inside the intermetallic phases. 

As a confirmation of the presented results with the literature [13,46], the “X” area, which showed the crack path in Figure 11, contained the Al or Mn elements as intermetallic phases. Most cracks were from Mg-Al intermetallic phases and one case was due to Mn. Notably, the Mg matrix alone was not the origin of cracks, based on its own effects, since it is softer than intermetallic phases.

7. Why were the different color SEM images displayed in figure 11? Any specific reason beyond this?

Answer: No, each element has its own color. For elements, Mg is in orange, Al is in purple, and Mn is in blue. There are no any other reasons for this type of presentation. The application of these colors is to used all elements in one image that is not presented here. 

Journal Requirements:

Answer: Thank you for the comments. Both procedures are considered for the revision. 

2. Thank you for stating the following financial disclosure: "No"

Answer: The last one is mentioned in the revised article. 

Answer: All references are again check to have complete details. All corrected ones were highlighted in yellow colors.

---

## [Decision Letter · Decision Letter 1]

19 Jan 2023

Amazing Epsilon-shaped trend for fretting fatigue characteristics in AM60 magnesium alloy under stress-controlled cyclic conditions at bending loads with zero mean stress

PONE-D-22-31978R1

Dear Dr. Azadi,

We’re pleased to inform you that your manuscript has been judged scientifically suitable for publication and will be formally accepted for publication once it meets all outstanding technical requirements.

Kind regards,

Khalil Abdelrazek Khalil, Ph.D.

Academic Editor

PLOS ONE

Additional Editor Comments (optional):

Reviewers' comments:

Reviewer's Responses to Questions

**Comments to the Author**

1. If the authors have adequately addressed your comments raised in a previous round of review and you feel that this manuscript is now acceptable for publication, you may indicate that here to bypass the “Comments to the Author” section, enter your conflict of interest statement in the “Confidential to Editor” section, and submit your "Accept" recommendation.

Reviewer #1: All comments have been addressed

Reviewer #2: All comments have been addressed

2. Is the manuscript technically sound, and do the data support the conclusions?

Reviewer #1: Yes

Reviewer #2: Yes

3. Has the statistical analysis been performed appropriately and rigorously? 

Reviewer #1: Yes

Reviewer #2: Yes

4. Have the authors made all data underlying the findings in their manuscript fully available?

Reviewer #1: Yes

Reviewer #2: Yes

5. Is the manuscript presented in an intelligible fashion and written in standard English?

Reviewer #1: Yes

Reviewer #2: Yes

6. Review Comments to the Author

Reviewer #1: No further revision is required, as the reviewer's comments have been appropriately addressed. Therefore, the manuscript is eligible for publication.

Reviewer #2: Everything has done perfectly. Formatting and figure quality should be improved. Table captions should be improved in the comprehensive manner.

7. PLOS authors have the option to publish the peer review history of their article (what does this mean?). If published, this will include your full peer review and any attached files.

Reviewer #1: No

Reviewer #2: No

---

## [Editor Report · Acceptance letter]

30 Jan 2023

PONE-D-22-31978R1 

Amazing Epsilon-shaped trend for fretting fatigue characteristics in AM60 magnesium alloy under stress-controlled cyclic conditions at bending loads with zero mean stress 

Dear Dr. Azadi:

I'm pleased to inform you that your manuscript has been deemed suitable for publication in PLOS ONE. Congratulations! Your manuscript is now with our production department. 

Kind regards, 

on behalf of

Dr. Khalil Abdelrazek Khalil 

Academic Editor

PLOS ONE